# Zero and Few-shot Semantic Parsing with Ambiguous Inputs

**Elias Stengel-Eskin**[1]    **Kyle Rawlins**[2]    **Benjamin Van Durme**[2]
[1]UNC Chapel Hill    [2]Johns Hopkins University

## Abstract

Despite the frequent challenges posed by ambiguity when representing meaning via natural language, it is often ignored or deliberately removed in tasks mapping language to formally-designed representations, which generally assume a one-to-one mapping between linguistic and formal representations. We attempt to address this shortcoming by introducing AMP, a framework, dataset, and challenge for translating ambiguous natural language to formal representations like logic and code. We define templates and generate data for five well-documented linguistic ambiguities. Using AMP, we investigate how several few-shot text-to-code systems handle ambiguity, introducing three new metrics. We find that large pre-trained models perform poorly at capturing the distribution of possible meanings without deliberate instruction. However, models are able to capture the distribution well when ambiguity is attested in their inputs. These results motivate a call for including ambiguity explicitly in datasets and promote considering the distribution of possible outputs when evaluating systems. [1]

## 1 Introduction

Formalizing the meaning of natural language into a symbolic representation has been attempted across a variety of domains, from philosophy and linguistics (Wittgenstein, 1921; Montague, 1970) to artificial intelligence (Winograd, 1972; Zelle & Mooney, 1996). Attempts at formalization have often faced a shared challenge: many natural language statements have multiple possible meanings, i.e. they are ambiguous. Past work (e.g. Zipf, 1949; Piantadosi et al., 2012) has argued that this is a natural feature of a communication system, resulting from competing pressures on speakers and listeners. Specifically, Piantadosi et al. contend that ambiguity allows speakers to minimize their efforts. Rather than exactly specify their intended meaning (resulting in a long and expensive message), speakers can send shorter, cheaper messages and rely on listeners to resolve any ambiguities. However, this resolution in turn relies on *commonsense knowledge* and *conversational context*: most speakers of English would infer from the utterance, *"I ate spaghetti with a fork"* that someone used a fork as a utensil, but commonsense knowledge would preclude this parse of *"I ate spaghetti with meatballs"*. Similarly, conversational context can provide clues to help us choose between interpretations.

Language can be used not only to communicate with other people, but also to interact with AI agents. One common method for interaction is semantic parsing, whereby natural language is translated into a formal and symbolic representation of its meaning (e.g. code, logic, graphs, etc.). However, human tools for ambiguity resolution may be unavailable to these non-human translation systems: models typically lack human-like commonsense knowledge and are missing conversational context. This could lead to miscommunications between humans and models. Since semantic parsing systems are used to perform real-world actions (e.g. modifying a calendar, sending emails, controlling physical robots, etc.) ambiguity-based miscommunication in parsing could have real-world consequences.

Ideally, given an ambiguous input, we would like our parsing models to capture a *distribution* over interpretations with some uncertainty across plausible items in the distribution. This would allow robust handling of ambiguous utterances – for example by enabling smart follow-up interactions (Stengel-Eskin & Van Durme, 2023) – getting us closer to the goal of using language as a general-purpose API for interaction. Given that semantic parsing systems are typically based on language models – which represent distributions over strings – combined with a search procedure (e.g. beam

---

[1]Data and code: `https://github.com/esteng/ambiguous_parsing`. Contact: `esteng@cs.unc.edu`

*The boy saw the man with the telescope*

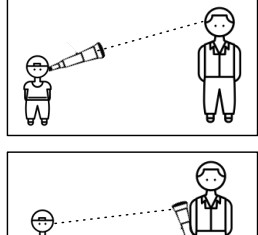

$\exists x.\exists y.\exists z.\exists a.boy(x) \wedge man(y)$
$\wedge\, telescope(z) \wedge saw(a) \wedge$
$agent(a, x) \wedge patient(a, y) \wedge$
$instrument(a, z)$

```
(exists x (exists y (exists z ( exists a
 (AND (boy x) (man y) (telescope z)
      (saw a) (agent a x) (patient a y)
      (instrument a z))))))
```

$\exists x.\exists y.\exists z.\exists a.\exists e.boy(x)$
$\wedge\, man(y) \wedge telescope(z)$
$\wedge\, saw(a) \wedge agent(a, x)$
$\wedge\, patient(a, y) \wedge have(e)$
$\wedge\, agent(e, x) \wedge patient(e, z)$

```
(exists x (exists y (exists z ( exists a
(exists e (
 (AND (boy x) (man y) (telescope z)
      (saw a) (agent a x) (patient a y)
      (have e) (agent e x) (patient e z)
)))))))
```

Figure 1: An example of prepositional phrase (PP) attachment ambiguity. The statement is compatible with two possible interpretations, represented visually, in first-order logic, and in Lisp format.

search) it could be that models already capture ambiguity. However, this hypothesis is hard to test given current semantic parsing datasets, which typically commit to a single interpretation for each utterance. To this end, we introduce an extensible framework and dataset for investigating ambiguity in semantic parsing. Our framework consists of templates covering five well-documented types of natural language ambiguity: prepositional-phrase attachment, scope and inverse scope, pronominal coreference, and conjunctions. For each type, our templates can generate large numbers of ambiguous and unambiguous utterances. Each ambiguous utterance is paired with two possible interpretations, or logical forms (LFs); see Fig. 1 for an example. LFs can be represented as first-order-logic (FOL) formulae or as programs in Lisp. We use our framework to create a benchmark dataset we call AMP (**Am**biguous **P**arsing). Unlike past efforts which have grounded ambiguous utterances in answers to questions (Stengel-Eskin et al., 2023), language inferences (Liu et al., 2023), videos (Berzak et al., 2015), or images (Mehrabi et al., 2022), we focus our dataset on semantic parses. This choice follows from several motivating factors. For one, semantic parsing has a long tradition of use in interactive systems, including in robotics (Kate et al., 2005; Tellex et al., 2011; Artzi & Zettlemoyer, 2013; Tellex et al., 2020), question-answering (Zelle & Mooney, 1996; Berant et al., 2013; Yu et al., 2018), and digital assistants (Semantic Machines et al., 2020; Damonte et al., 2019). Ensuring that these systems capture ambiguity and that their confidence reflects appropriate uncertainty about the user's intent is crucial, as misunderstandings could have negative real-world consequences (Stengel-Eskin & Van Durme, 2023). Furthermore, semantic parsing not only allows people to access computation, but also provides a way for models to use external tools: for example, simple forms of semantic parsing have been employed to augment large language models (LLMs) (Parisi et al., 2022; Schick et al., 2023; Mialon et al., 2023). Finally, long-form text-to-code shares many challenges with parsing.

Using our generated AMP data, we introduce a pair of challenging tasks designed for LLMs using in-context learning (ICL). In ICL, rather than explicitly training models to predict LFs, we provide models with instructive examples in a prompt, which is prepended to the test input. This parsing setting has become increasingly popular in semantic parsing (Shin et al., 2021; Shin & Van Durme, 2022; Roy et al., 2022). Our tasks aim to quantify how well existing models capture ambiguity and to provide a framework for improving their ability to predict multiple meanings. We develop 3 metrics to measure models' performance on ambiguity in two settings: **zero-shot** and **mixed prompt**.

In the **zero-shot** setting, we provide models with the "ingredients" to produce both possible derivations of a given ambiguity type, but we provide no examples of that ambiguity type; see Appendix A.2 for an example. In this unique compositional generalization challenge, the model must combine structures into novel derivations and *also* recognize that the structures can be selected and combined in two ways to produce different derivations. We also annotate a subset of our data with crowdsourced judgements, comparing these to our models' predictions. Models struggle to predict parses correctly in this setting. When they do compose parses correctly, although models and people tend to choose similar interpretations, models generally fail to predict both possible parses.

In the **mixed prompt** setting, we examine how model distributions and outputs change when varying the number of examples for each interpretation in the prompt. For each ambiguity type, we construct "mixed prompts" consisting of conflicting examples. Some examples shown to the model pair utterances of an ambiguity type with one kind of LF, and others pair the same kinds of inputs with the alternative LF. This setting is motivated by a case in which ambiguity might lead to conflicting annotations in a training dataset; when examples are retrieved from that data to construct a prompt for

| Type | Ex. Input | $LF_0$ | $LF_1$ |
|---|---|---|---|
| Prep. phrase attachment (PP) | *The man saw the boy with the telescope* | $\exists x.\exists y.\exists z.\exists a.\exists e.man(x) \wedge boy(y) \wedge saw(a) \wedge agent(a,x) \wedge patient(a,y) \wedge telescope(z) \wedge have(e) \wedge agent(e,y) \wedge patient(e,z)$ *Interpretation*: the man saw the boy, who was holding a telescope. | $\exists x.\exists y.\exists z.\exists a.man(x) \wedge boy(y) \wedge telescope(z) \wedge saw(a) \wedge agent(a,x) \wedge patient(a,y) \wedge instrument(a,z)$ *Interpretation*: the man used a telescope to see the boy. |
| Quantifier scope (Scope) | *every cow saw a dog* | $\exists x.\forall y.\exists a.cow(y) \wedge dog(x) \wedge saw(a) \wedge agent(a,y) \wedge patient(a,x)$ *Interpretation*: there is *exactly* one dog. | $\forall x.\exists y.\exists a.cow(x) \wedge dog(y) \wedge saw(a) \wedge agent(a,x) \wedge patient(a,y)$ *Interpretation*: there may be more than one dog. |
| Reversed, or inverse scope (revs-cope) | *a cow saw every dog* | $\exists x.\forall y.\exists a.cow(x) \wedge dog(y) \wedge saw(a) \wedge agent(a,x) \wedge patient(a,y)$ *Interpretation*: there is *exactly* one cow. | $\forall x.\exists y.\exists a.cow(y) \wedge dog(x) \wedge saw(a) \wedge agent(a,y) \wedge patient(a,x)$ *Interpretation*: there may be more than one cow. |
| Pronoun coreference (bound) | *Mary saw the woman and she smiled* | $\exists x.\exists a.\exists e.woman(x) \wedge saw(a) \wedge agent(a,Mary) \wedge patient(a,x) \wedge smiled(e) \wedge agent(e,Mary)$ *Interpretation*: Mary smiled. | $\exists x.\exists a.\exists e.woman(x) \wedge saw(a) \wedge agent(a,Mary) \wedge patient(a,x) \wedge smiled(e) \wedge agent(e,x)$ *Interpretation*: the woman smiled. |
| Conjunction (conj.) | *the man drank and ate or swam* | $\exists x.\exists a.\exists e.\exists i.man(x) \wedge ((drank(a) \wedge agent(a,x) \wedge ate(e) \wedge agent(e,x)) \vee (swam(i) \wedge agent(i,x)))$ *Interpretation*: the man either drank and ate or he swam. | $\exists x.\exists a.\exists e.\exists i.man(x) \wedge (drank(a) \wedge agent(a,x) \wedge ((ate(e) \wedge agent(e,x)) \vee (swam(i) \wedge agent(i,x))))$ *Interpretation*: the man drank, and he either ate or swam. |

Table 1: Ambiguity types considered with example inputs and LFs. See Appendix A.1.1 for more description, including the lexical items used.

a test example, the resulting prompt will also contain conflicting parses. Here, our metrics measure to what extent a model represents the distribution in its input given conflicting evidence. Some models perform remarkably well here, aligning with the prompt distribution across ambiguity types. To our knowledge, this is the first study of in-context learning with conflicting evidence.

## 2 METHODS

**Data** We introduce a dataset of ambiguous parses, where natural language examples are parsed into first-order logic (FOL). Further details on the construction of our logical forms (LFs) can be found in Appendix A.1. We can canonicalize our LFs, so that logically equivalent formulae with varying syntax are treated as identical: we transform LFs into binary trees, where nodes are ordered alphabetically, and we anonymize variables. Note that when prompting our model, we do use a standard variable set and order, where the variables $x, y, z$ are used for nouns, and $a, e, i$ are used for events. Our canonicalization process also allows us to render our LFs in different formats. In addition to a standard FOL format, we experiment with a Lisp format (cf. Fig. 1). For machine-readability, we always render logical connectives in plaintext, i.e. $\exists$ becomes `exists`, $\wedge$ becomes `AND`, etc.. We consider five types of syntactic and semantic ambiguities, given in Table 1.[2]

**Models** Semantic parsing tasks are often framed as sequence transduction, where a model learns to translate text into LFs by training on paired data (Dong & Lapata, 2016; Zhang et al., 2019). It has become clear that neural models can capture distributions they are trained on; thus, if we were to train on ambiguous data, it would not be surprising if the model captured ambiguity, and vice-versa. Rather than training models, we instead consider several models for in-context learning (ICL), focusing on large pre-trained autoregressive (AR) language models. We use the Codegen series of models (Nijkamp et al., 2022) – 350 million (M), 2 billion (B), 6B, and 16B parameters – which are based on the GPT-2 architecture (Radford et al., 2019) and are pre-trained on large amounts of code and text.[3] We also use LLMs pre-trained on text; here, we examine Llama-13B (Touvron et al., 2023), an open-source AR transformer. To examine the impact of instruction tuning (Wei et al., 2022), we consider Vicuna-13B (Chiang et al., 2023), which uses prompts distilled from ChatGPT to

---

[2]We can also generate unambiguous examples, and can extend AMP to new ambiguity types/vocab items.

[3]Past work (Shin & Van Durme, 2022) has shown that code pretraining improves over text pre-training on other ICL semantic parsing tasks.

instruction-finetune Llama-13B. All models above 350M were run at `fp16` precision. In the zero-shot setting, we also consider OpenAI's gpt-3.5-turbo. While few details about the model are known, it is a large AR transformer model which has undergone both instruction tuning and fine-tuning from human feedback (Ouyang et al., 2022). It is often the most performant model; however, the API does not provide access to logit scores, precluding analyses of uncertainty. As such, we only use it in our zero-shot experiments, where the metric is accuracy-based rather than uncertainty-based. For non-API models, we used constrained decoding (Shin et al., 2021; Shin & Van Durme, 2022; Roy et al., 2022) to ensure the model only produces valid logical statements; see Appendix A.3 for details.

**Computing probability under a forced decode**    In our analyses, we would like to compare the probabilities the model assigns to $LF_0$ and $LF_1$. While one could compare the product of probabilities under the model for each LF, we find that in practice, this results in very low scores for either LF. We instead use Stengel-Eskin & Van Durme (2022)'s sequence confidence estimate to obtain $P_\theta(LF_0)$, renormalizing at the end:

1. We obtain token-level probabilities under a forced decode of $LF_0$ and $LF_1$. For each token $t_i$ in an LF with tokens $t_1 \ldots t_N$, we compute $P_\theta(t_i|x; t_{1:i-1})$, where $t_{1:i-1}$ is the *gold* token prefix and $x$ is the input prompt.
2. We take $\hat{p} = \min_{i=1}^{N} P_\theta(t_i|x; t_{1:i-1})$, the minimum probability across all tokens.[4]
3. We normalize the probabilities: $P_\theta(LF_0) = \frac{\hat{p}_{LF_0}}{\hat{p}_{LF_0} + \hat{p}_{LF_1}}$ and set $P_\theta(LF_1) = 1 - P_\theta(LF_0)$

## 2.1 METRICS

**Zero-shot metrics**    In the zero-shot setting, we aim to measure the degree to which the model captures both possible interpretations of an ambiguous statement. Intuitively, when given the "ingredients" to make both interpretations, a model that robustly captures ambiguity should allocate some probability to both. We measure this by computing the proportion of elements for which the model has both interpretations in its top-$k$ predictions. Note that we remain agnostic here to the exact probability of each interpretation; we aim instead to quantify whether it predicts both interpretations at all. Note also that as we increase $k$, this metric becomes less stringent. Let $T_k$ be the top $k$ most probable predictions from the model under some sampling method (e.g. beam search), and $\mathbb{I}$ be an indicator function. The zero-shot metric $ZM_k$ is given by Eq. (1). This metric counts how often both LFs are found in the top $k$ outputs averaged across examples $i \in [1, N]$.. It ranges from 0 to 100, and higher is better, as it indicates more examples have both LFs in their top $k$ outputs.

$$ZM_k = \frac{\sum_{i=1}^{N} \left( \mathbb{I}[LF_0 \in T_k] * \mathbb{I}[LF_1 \in T_k] \right)}{N} * 100 \tag{1}$$

**Few-shot metrics**    In the few-shot setting, we are concerned about the level to which the model is capturing the distribution given in the prompt. A core assumption here is that an ideal model would perfectly capture the uncertainty in the given distribution. We consider metrics at two levels of granularity to evaluate this behavior. The first metric we consider measures model performance at the level of the dataset. Intuitively, as we sweep across ratios $r \in R$, we expect the proportion of predicted LFs to match $r$. For example, when $r = 0.10$ (meaning that 10% of the prompt examples are $LF_0$ and 90% are $LF_1$) we would expect the model to produce $LF_0$ in roughly 10% of instances. Let $y_i$ be the predicted LF for input instance $x_i$. Then the fewshot dataset metric $FDM$ is given by Eq. (2). Intuitively, this measures the difference between the accuracy on each LF and the ratio of that LF; lower is better for $FDM$, which ranges from 1.0 to 0.0. The second metric measures model performance at the level of individual datapoints. If the model is capturing the distribution in the prompt, then the probability assigned to $LF_0$ should roughly match $r$, e.g. if $r = 0.10$, the model should assign $P(LF_0) \approx 0.10$. The few-shot instance metric $FIM$ is given by Eq. (3). $FIM$ resembles a Brier score (Brier et al., 1950) and measures the error between the predicted probability and the ratio; it also ranges from 1.0 to 0.0 and lower is better.

$$FDM = \frac{1}{|R|} \sum_{r \in R} \left( \left| \left( \frac{1}{N} \sum_{i=1}^{N} \mathbb{I}[y_i = LF_0] \right) - r \right| + \left| \left( \frac{1}{N} \sum_{i=1}^{N} \mathbb{I}[y_i = LF_1] \right) - (1-r) \right| \right) \tag{2}$$

$$FIM = \frac{1}{|R|} \sum_{r \in R} \left( \frac{1}{N} \sum_{i=1}^{N} (P_\theta(y_i = LF_0) - r)^2 \right) \tag{3}$$

---

[4]We also experimented with averaging, which resulted in similar results. We chose min to be consistent with Stengel-Eskin & Van Durme (2022).

## 3 Experiment 1: Zero-shot parsing

For each ambiguity type, we construct a prompt that provides the ingredients for deriving both LFs. The order of the component sentences is shuffled to avoid biasing the model towards one interpretation or the other. Crucially, the prompt contains no examples of the types of sentences being tested. For example, for PP attachment, the model is given an example of how to parse transitive verbs (*"the boy saw the man"*), instruments (*"the boy saw with the telescope"*), and possessives (*"the boy with the telescope"*), each in isolation. To successfully generalize, the model has to overcome two challenges: first, it must compositionally generalize to compose the ingredients in the prompt into a valid derivation. Secondly, it must recognize the ambiguity and reflect both derivations in its output. For each ambiguity type, we test 200 examples. Prompt examples are given in Appendix A.2.

### 3.1 Zero-shot results and analysis

In Fig. 2, smaller Codegen models (350M, 2B) struggle to predict either LF correctly. On some ambiguity types, larger Codegen models (6B, 16B) predict one LF correctly. However, most models fail to ever predict both LFs correctly; exceptions to this are conjunction and coreference ambiguities, where we see some models correctly predicting $LF_0$ for some examples and $LF_1$ for others. GPT-3.5 does well at predicting $LF_1$ for PP and scope ambiguities, and predicts both LFs for coreference and conjunctions. Interestingly, while Llama-13B is unable to correctly predict either LF for any of the ambiguities, Vicuna-13B (Llama's instruction-tuned variant) is comparable to Codegen-16B, suggesting that instruction tuning helps the model predict one LF correctly (though not to capture ambiguity). Separately, we find that predicting FOL generally outperforms Lisp; for example, the Codegen-2B model on scope predicts $LF_1$ correctly $18\%$ of the time when using FOL and only $11\%$ when using Lisp; we report only FOL results moving forward.

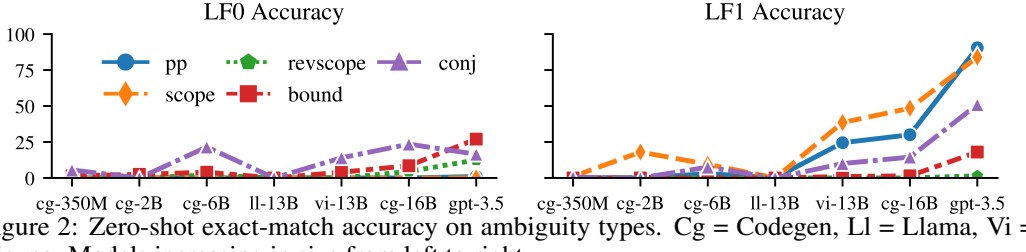

Figure 2: Zero-shot exact-match accuracy on ambiguity types. Cg = Codegen, Ll = Llama, Vi = Vicuna. Models increasing in size from left to right.

These results are further underscored in Table 2 showing the $ZM_5$ values for all models. We see that all models tested perform very poorly on this metric, with most models scoring $0.0$ in most settings. Qualitatively, we find that the model's top 5 outputs tend to include variations of the same LF. This finding aligns with the probability results seen in Fig. 3, where models tend to assign extreme probabilities to LFs on 3 out of 5 ambiguity types. Notable exceptions here are conjunction and bound pronoun types, where in Fig. 3 models assign closer to 0.5 probability to each parse; we see this also reflected in Fig. 2 and Table 2, where models predict both $LF_0$ and $LF_1$ correctly some of the time. As a whole, these results underscore the difficulty of the compositional task we have proposed; while some models are able to obtain high accuracy on one LF in isolation (GPT-3.5 predicts $LF_1$ for PP attachment and scope almost perfectly) no model is able to consistently predict both interpretations.

We can also ask whether token-level confidence reflects the ambiguity in the space of possible parses. Examining task-oriented semantic parsing models, Stengel-Eskin & Van Durme (2022) find that many models (including Codegen) are relatively well-calibrated at the token level, meaning their confidence aligns with their average accuracy. Taking token-level probabilities as confidence scores, we follow their analysis and ask whether models are well-calibrated w.r.t. alternative parses. Specifically, we compare the model's confidence on tokens at the points where the predicted and alternative parse diverge. This is visualized for each ambiguity type in Fig. 4. Here, we take the first correctly-predicted LF (either $LF_0$ or $LF_1$) from Codegen-16B, predicted via beam search with grammar-constrained decoding (not forced decoding). We overlay the confidence onto the token as the background color (darker is more confident). Below each predicted parse, we give the alternative parse. For scope and inverse scope, the model assigns low confidence to the quantifier tokens at the start of the formula, which are in the reverse order in the alternative parse. Similarly, the tokens involving the quantified variables have lower confidence. We also see low confidence around the area of divergence for conjunction. However, pronominal coreference and PP attachment lack such

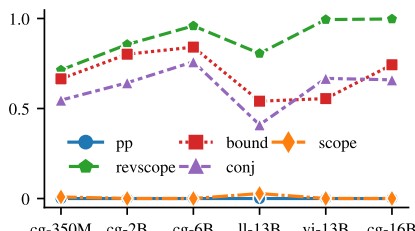

Figure 3: $P(LF_0)$ per model.

| Model | PP | Scope | Revscope | Bound | Conj. |
|---|---|---|---|---|---|
| cg-350M | 0.00 | 0.00 | 0.00 | 0.00 | 1.00 |
| cg-2B | 0.00 | 0.00 | 0.00 | 0.50 | 0.00 |
| cg-6B | 0.00 | 0.00 | 0.00 | 0.00 | 3.50 |
| cg-16B | 0.00 | 0.00 | 0.00 | 3.50 | 15.00 |
| ll-13B | 0.00 | 0.00 | 0.00 | 0.00 | 0.00 |
| vi-13B | 0.00 | 0.00 | 0.00 | 4.00 | 9.50 |
| gpt-3.5 | 0.00 | 0.00 | 0.00 | 0.00 | 0.00 |

Table 2: $ZM_5$ for all models (cg = Codegen). Models generally fail to predict both LFs.

scope (a): forall x . exists y . exists a . boy(x) AND pyjamas(y) AND observed(a) AND agent(a, x) AND patient(a, y)

scope (b): exists x . forall y . exists a . boy(y) AND pyjamas(x) AND observed(a) AND agent(a, y) AND patient(a, x)

revscope (a): exists x . forall y . exists a . boy(x) AND sweater(y) AND spied(a) AND agent(a, x) AND patient(a, y)

revscope (b): forall x . exists y . exists a . boy(y) AND sweater(x) AND spied(a) AND agent(a, y) AND patient(a, x)

bound (a): exists x . exists a . exists e . girl(x) AND spied(a) AND agent(a, x) AND patient(a, Mary) AND smiled(e) AND agent(e, x)

bound (b): exists x . exists a . exists e . girl(x) AND spied(a) AND agent(a, x) AND patient(a, Mary) AND smiled(e) AND agent(e, Mary)

conj (a): exists x . exists a . exists e . exists i . cat(x) AND ( ( drank(a) AND agent(a, x) ) OR ( ate(e) AND agent(e, x) ) ) AND ( played(i) AND agent(i, x) )

conj (b): exists x . exists a . exists e . exists i . cat(x) AND ( ( drank(a) AND agent(a, x) ) OR ( ate(e) AND agent(e, x) AND played(i) AND agent(i, x) ) )

pp (a): exists x . exists a . camera(x) AND saw(a) AND agent(a, Watson) AND instrument(a, x) AND patient(a, Galileo)

pp (b): exists x . exists a . exists e . camera(x) AND saw(a) AND agent(a, Watson) AND patient(a, Galileo) AND have(e) AND agent(e, Galileo) AND patient(e, x)

Figure 4: Zero-shot per-token probability (darker is more probable) for each ambiguity type. Alternative parse given below each predicted parse. Token probability sometimes reflects divergences between the parses.

interpretable confidence changes. These results are promising: for some ambiguity types, the model's token-level confidence reflects the alternative parse.

## 3.2 HUMAN VALIDATION

Fig. 2 indicates that models tend to produce one interpretation or the other – when models have non-zero accuracy on one interpretation, they tend to have zero on the other. Psycholinguistic research suggests that people have preferred interpretations (AnderBois et al., 2012; Dwivedi, 2013); at the aggregate level, Fig. 3 shows that models align with human preferences on scope ambiguities.

AnderBois et al. (2012) and Dwivedi (2013) also describe strong lexical effects in scope ambiguity, meaning that the choice of words in the example has an effect on the interpretation taken. In order to further examine how the models tested compare with these results, we annotate a subset of our validation examples with human interpretations and confidence scores. This allows us to compare model predictions to humans at an item-level in addition to an aggregate level.

Annotators were asked to choose between interpretations and provide a confidence score on a sliding scale, following the EASL protocol (Sakaguchi & Van Durme, 2018); the confidence score was then converted to a probability (cf. Appendix A.5 for details). Since annotators are unlikely to know FOL or Lisp, each LF is shown as a statement that clearly indicates the interpretation (as in Table 1). For example, for a PP attachment example like, *"the boy saw the man with the telescope"*, the verbalized interpretations are *"the boy saw the man, who had/was wearing a telescope"* and *"the boy saw the man and used a telescope to do so"*. For each ambiguity type except conjunctions, we randomly select 20 examples from our development splits.[5] Each example is annotated by 3 annotators.

We find that annotators disagree almost as often as they agree: 38 examples have disagreement while 42 have all 3 annotators agreeing. This is a positive finding, indicating that our examples are highly ambiguous. Fig. 5 (left) shows confidence scores (averaged across 3 annotators) for each item (sorted separately by mean confidence for each ambiguity type). There are broad preferences for all ambiguity types except PP attachment: bound and inverse scope tend to be matched to $LF_0$, and scope to $LF_1$. The latter aligns with some past findings AnderBois et al. (2012); Caramazza et al. (1977).[6] For PP attachment, some inputs are confidently parsed as $LF_0$ and others as $LF_1$.

In Fig. 5 (right), we contrast the human results with the output of the Codegen-16B model. While the probabilities generally match in direction to the human annotations (except for PP-attachment) we

---

[5]Conjunction ambiguities were excluded due to difficulties in verbalizing their interpretations fluently.

[6]Other work has found linear order to have a negligible effect and pointed to additional factors influencing interpretations (Kurtzman & MacDonald, 1993; Dwivedi, 2013).

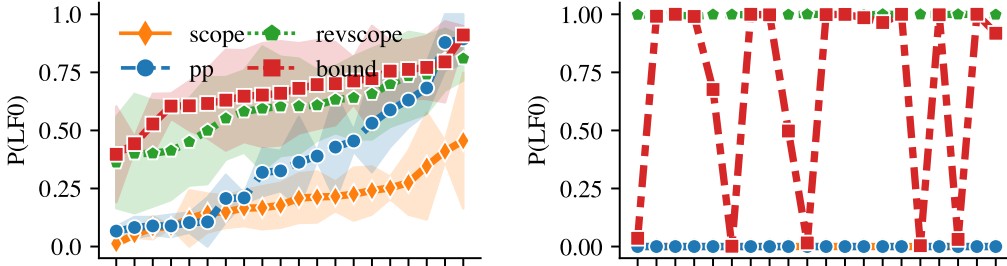

Figure 5: Per-example probabilities derived from humans (left) and cg-16B (right) on LFs. Examples are sorted by probability. Human probabilities vary according to vocabulary choice, but model probabilities generally do not.

do not see the same kind of item-level sensitivity. For most PP, scope, and inverse scope examples, we see the model assigning all examples the same extreme probability. For bound pronouns, we see more variation, with the model switching between $LF_0$ and $LF_1$; however, the predictions are fairly extreme. These results suggest that the model is poorly-calibrated w.r.t. ambiguity at the item level.

## 4 EXPERIMENT 2: FEW-SHOT PARSING

Ambiguities may lead to similar inputs being paired with different logical forms. Increasingly, it is common to retrieve examples from a training set to compose a prompt for ICL. If that training set has ambiguity in it, it is likely that the retrieved prompt would contain conflicting examples, e.g. some examples pairing an utterance type with $LF_0$ and others pairing it with $LF_1$. In our few-shot experiments, we seek to fill this gap by investigating how model confidence and accuracy change at different prompt ratios. Crucially, by investigating mixed prompts with ambiguous inputs, we are ensuring that the disagreement in the prompt is not due to simple mistakes; one could imagine a mixed prompt arising from noisy data, where instances are mislabeled. In such cases, a strong enough model may even learn to ignore mislabeled data in the prompt. However, in the case of ambiguity, there are multiple *legitimate* interpretations.

For each ambiguity type, we construct prompts by pairing sentences of the same type with $LF_0$ in some cases, and $LF_1$ in others; we run 100 examples per type, per ratio. Each prompt contains 10 input-LF pairs, and a different prompt is constructed for each test sentence. We vary the number of $LF_0$ sentences in the prompt from 0 to 10 in increments of 1 (e.g. $0 - 100\%$) and shuffle the prompt sentences to ensure that there is no positional bias.

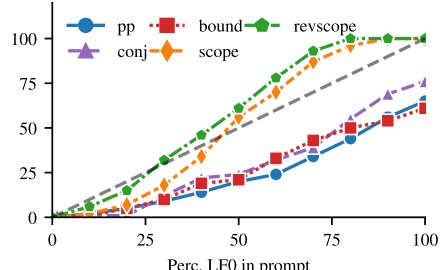

Figure 6: Fewshot acc. increases according to the ratio of the LF in the prompt. $LF_1$ acc. (not pictured) decreases accordingly.

**Few-shot results and analysis** Fig. 6 shows the accuracy of models on $LF_0$ as we increase the percentage of $LF_0$ in the prompt for the Codegen-2B model. We see that for scope and inverse scope, the accuracy tracks almost perfectly with the percentage. For other ambiguities, the accuracy is correlated with the percentage but never reaches $100\%$. Table 3 shows the $FIM$ and $FDM$ scores for all models across all ambiguity types. Recall that $FDM$ measures how well the model's *accuracy* aligns with the percentage of examples for each LF in the prompt: to obtain a lower $FDM$ score, the model needs to predict each LF at roughly the rate that it is seen in the prompt. Note that a model that fails to predict either LF correctly will have a high $FDM$ score. For example, because Llama-13B fails to predict any of the LFs correctly, it has an $FDM$ score of 1.00 for all ambiguities. For Codegen models, $FDM$ generally improves with model size on most ambiguity types. Overall, several models achieve fairly low $FDM$, especially on scope and inverse scope.

We also see that all $FIM$ scores are relatively low. Because $FIM$ uses the gold LF to extract the sequence probability, even models with poor accuracy on both LFs can have a fairly low $FIM$. In other words, $FIM$ presents the model with a forced choice between two parses, rather than evaluating the model's most probable generations like $FDM$ and $ZM$ do. For example, Llama-13B never produces a correct LF under beam-search decoding with constraints, but when probabilities are

extracted with a forced decode of the gold LFs, they align fairly well with those in the prompt, and it often attains lower $FIM$ scores than other models, including Vicuna. $FIM$ remains fairly constant with model size: Codegen-16B is tied with smaller models on 4/5 ambiguity types. Taken together, these results indicates that the models tested are surprisingly good at capturing the distribution in the prompt. The low $FDM$ on some types indicates that overall, models like Codegen-16B and Vicuna-13B produce $LF_0$ roughly at the rate that it appears in the prompt. However, we see that low $FIM$ does not imply low $FDM$, since low $FDM$ requires the model to be accurate.

Interestingly, the models seem to override the zero-shot tendencies seen in Fig. 3, where scope and inverse scope were strongly associated with one LF over the other. With direct evidence on how to parse scope sentences in the prompt – evidence we did not provide in the zero-shot setting – the model produces the interpretations seen in the prompt, and is especially close to the prompt distribution for scope and inverse scope. These results are promising: given that models seem to capture mixed prompts well, it could be that ambiguity poses less of a challenge in settings where such mixed prompts can be constructed, i.e. settings with ambiguity in the training data.

| Model | PP | | Scope | | Revscope | | Bound | | Conj. | |
|---|---|---|---|---|---|---|---|---|---|---|
| – | $FDM$ | $FIM$ | $FDM$ | $FIM$ | $FDM$ | $FIM$ | $FDM$ | $FIM$ | $FDM$ | $FIM$ |
| cg-350M | 0.76 | 0.08 | 0.19 | 0.05 | 0.36 | 0.06 | 0.62 | 0.16 | 0.60 | 0.06 |
| cg-2B | 0.51 | 0.09 | 0.19 | 0.03 | 0.18 | 0.03 | 0.48 | 0.10 | 0.39 | 0.05 |
| cg-6B | 0.45 | 0.08 | 0.21 | 0.05 | 0.16 | 0.06 | 0.43 | 0.10 | 0.41 | 0.06 |
| cg-16B | 0.35 | 0.08 | 0.20 | 0.03 | 0.21 | 0.03 | 0.38 | 0.10 | 0.27 | 0.06 |
| ll-13B | 1.00 | 0.06 | 1.00 | 0.04 | 1.00 | 0.04 | 1.00 | 0.06 | 1.00 | 0.05 |
| vi-13B | 0.50 | 0.08 | 0.17 | 0.05 | 0.20 | 0.06 | 0.26 | 0.07 | 0.35 | 0.09 |

Table 3: Few-shot metrics for all models (lower=better). $FDM$ (Eq. (2)) measures the extent to which the model's accuracy across the whole dataset matches the percentage of that LF in the prompt. $FIM$ (Eq. (3)) measures how well the model's uncertainty captures the prompt's uncertainty.

## 5 RELATED WORK

Ambiguity has been a longstanding topic of interest in linguistics and psycholinguistics. Past work has argued that it is a feature arising naturally from the trade-off between competing objectives (Zipf, 1949; Schutze, 1995; Piantadosi et al., 2012). However, providing a systematic account of ambiguity in linguistics lies beyond the scope of this section.

Some work in NLP has focused on modeling ambiguity in visual contexts, where questions and statements have been paired with images or videos depicting situations they refer to. Stengel-Eskin et al. (2023) introduce a dataset of linguistically ambiguous questions about images as well as a model for question disambiguation, and Futeral et al. (2022) examine ambiguous source sentences in machine translation, providing disambiguating images. More akin to our work, Berzak et al. (2015) introduce a corpus of syntactic, semantic, and pragmatic ambiguities with video interpretations. Follow-up work by Mehrabi et al. (2022) generates images of ambiguous statements. We use many of the same ambiguities, but represent meaning with LFs instead of videos or images. This is motivated in part by the relative ease of checking the correctness of a logical formula over, for example, an image. Ambiguity has been studied in more general tasks such as question-answering (Min et al., 2020), natural language inference (NLI) (Liu et al., 2023), and coreference resolution (Yuan et al., 2023), where models have broadly been found lacking in their ability to resolve ambiguities. In parsing specifically, Rasmussen & Schuler (2020) introduce a $\lambda$-calculus dataset on 2,000 sentences of simple Wikipedia text, where roughly 50% contain quantifier scope ambiguity. We use synthetic data instead, giving us greater control and allowing us to examine more ambiguity types.

## 6 DISCUSSION AND CONCLUSION

In addition to semantic parsing's many downstream applications, it has often been used to measure models' compositional generalization abilities through synthetic benchmarks like COGS (Kim & Linzen, 2020) and SCAN (Lake & Baroni, 2018). The ability to generalize systematically and compositionally to unseen combinations is a core component of human intelligence (Fodor & Pylyshyn, 1988). Past efforts have generally assumed that there is a single correct LF for any given input, either implicitly (Lake & Baroni, 2018) or explicitly (Kim & Linzen, 2020, Appendix H). This assumption is not borne out in natural language, where statements can be ambiguous and have multiple meanings. It is also violated in many common applications of semantic parsing, such as

text-to-code, where there are myriad ways of producing logically equivalent programs. Making this assumption reduces semantic parsing to syntactic parsing, since there is a 1-to-1 mapping between syntax and meaning.[7] In future work, we hope to improve on the challenging and novel compositional task proposed in Section 3, where all models struggle to capture both meanings.

Section 4 offers a more hopeful takeaway: when ambiguity is present in the input, many models are able to capture the distribution of LFs. Of course, for ambiguity to be attested in the prompt, it must exist in the data used to construct the prompt. Furthermore, it needs to be attested in the evaluation data for us to test for it. However, in most current datasets, parses are not annotated redundantly or exhaustively, i.e. inputs are paired with a single output. Given that annotators often disagree on ambiguous examples (cf. Section 3.2) it is crucial to obtain multiple judgements, at least on evaluation data.[8] This has recently become more common in other domains, such as NLI (Chen et al., 2020; Nie et al., 2020; Pavlick & Kwiatkowski, 2019). Even when items are annotated redundantly, disagreement has often been discouraged or treated as noise. More recent work has begun to recognize that disagreement can arise for valid reasons (Pavlick & Kwiatkowski, 2019) including ambiguity (Bhattacharya et al., 2019; Stengel-Eskin et al., 2023; Liu et al., 2023). To improve the handling of ambiguity, we advocate for extending redundant, ambiguity-aware annotation protocols (with attention to disagreement) from single-label tasks (e.g. QA, NLI) to complex, sequential outputs like semantic parsing. Improving both zero-shot generalization and data collection would help models capture the full range of utterance interpretations. This could lead to robust, interactive systems in which agents ask for confirmation or clarification on ambiguous examples (Stengel-Eskin & Van Durme, 2023), ultimately improving safety for critical systems.

Finally, ambiguous utterances are underspecified, i.e. they lack the requisite information to decide which interpretation is correct. Some past work carries underspecification into the meaning representation: Copestake et al. (2005) introduces Minimum Recursion Semantics, which leaves noun-phrase bracketing (similar to conjunction ambiguities) and scope underspecified in the target representation, allowing multiple interpretations to be recovered. Bos (2004) introduce a discourse representation that also maintain scope ambiguities for later resolution. We have opted for a more fully-specified representation, placing the onus of resolution onto the parsing model rather than the representation.

**Limitations** Firstly, we are limited by committing to a particular logical form. To test the parsing abilities of models under ambiguity, we are forced to choose a fixed meaning representation (MR) form, possibly including suboptimal abstractions and design choices. Motivated by Wu et al. (2023), who find that arbitrary choices in an MR's construction can hamper compositional generalization, we have limited the number and difficulty of our choices, and mitigated their effect by offering two output formats. It is also important to point out some key differences between the models we test and the relevant psycholinguistic literature. Broadly, experiments indicate that people maintain multiple interpretations during *online* processing, later settling on one interpretation (Lackner & Garrett, 1972; Rayner et al., 1983; Filik et al., 2004). Our models do not do online processing, and receive input as text, not audio. Similarly, we do not provide a conversational context, which people generally use to resolve ambiguity. Thus, our results should not be taken to reflect how people process language.

Methodologically, we are limited by our use of a fixed set of English-only ambiguities. We hope to add more languages, lexical items, and ambiguity types via our extensible data framework. Many optimizations have could be made for ambiguity, e.g. decoding strategies that emphasize diversity (temperature sampling, sequentially decoding outputs, etc.) might result in better results for the $ZM$ metric, since models lack output diversity. In our experiments we attempted to mimic how semantic parsing is commonly done in practice, without optimizing for ambiguity specifically.

**Conclusion** By introducing a new benchmark for parsing under ambiguity, we are able to examine how modern semantic parsers handle cases where utterances can have multiple meanings. To this end, we introduced three new metrics for measuring the extent to which models capture the *distribution* of meanings. While we find that models struggle to compose symbols without explicit guidance, we also find that they are sensitive the ambiguity when given mixed prompts, suggesting that having ambiguity in the training data may be a sufficient condition for capturing it in the output. This motivates our call for capturing ambiguity during annotation.

---

[7]For example, Rudinger & Van Durme (2014) found that one of the key features separating dependency-based syntax and event-based semantics was the ability to handle PP attachment ambiguities.

[8]For training data, we may be able to obtain *diverse* judgements instead, with examples of a given ambiguity *type* paired with different outputs, as was done in Section 4

## 7 Ethics Statement

By creating sentences templatically, we ensure that AMP does not contain any harmful texts. This protects our human annotators from exposure to such inputs; we also compensate the annotators at a level substantially above US federal minimum wage, and in-line with living wage estimates. Like most NLP research, our work has the potential to contribute to dual-use. However, we believe that overall, making models robust to ambiguity will contribute to safer and more reliable technology, and has limited potential for negative applications.

## 8 Reproducibility Statement

In order to further reproducibility, we release our dataset and our code. This includes the code for AMP, which can be extended to generate new ambiguities, as well as the code for running all experiments. We do make use of a closed-source model (ChatGPT), which can hinder reproducibility; to hedge against this, most of our results are based on open-source models that are widely available. We also include our prompts in the appendix.

## Acknowledgements

This work has been supported in part by the U.S. National Science Foundation under grant No. 2204926. Elias Stengel-Eskin was supported by an National Science Foundation Graduate Research Fellowship.

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

# A APPENDIX

## A.1 DATASET CONSTRUCTION

We take a neo-Davidsonian event semantics approach (Parsons, 1990) to our logical forms (LFs), expressing our logical forms in quantified first-order logic (FOL). Events are represented as variables, with event-type predicates applied to them. For example, the statement *a woman walks* would be represented as $\exists x.\exists e.woman(x) \wedge walk(e) \wedge agent(e, x)$, allowing for an arbitrary number of semantic roles; the semantic roles covered in our dataset are *agent*, *patient*, and *instrument*. Generic noun phrases like *"a dog"* are existentially quantified: $\exists x.dog(x)$. Proper nouns are assumed to have a single referent, and are not quantified, e.g. *Mary walks* $\rightarrow \exists e.walks(e) \wedge agent(e, Mary)$.

### A.1.1 LEXICAL ITEMS

For PP ambiguity, we pair visual verbs (e.g. *see, observe, spot, etc.*) with visual instruments (e.g. *telescope, binoculars, etc.*) and tactile verbs (e.g. *grab, pick up, etc.*) with things that can be worn/possessed and used for manipulation (e.g. *gloves, ovenmitts, tongs, etc.*). For scope and reverse scope, we use common nouns and visual and tactile verbs. For pronoun coreference, the lexical items used here are gendered names (e.g *Mary, John, etc.*) and gendered nouns (e.g. *woman, man, boy, girl*). Conjunction examples use intransitive verbs.

### A.1.2 EXISTENTIAL QUANTIFICATION

For uses of *"the"*, we differ from Kim & Linzen (2020), who use the $\iota$ notation for definite articles to denote a uniqueness clause. This is implemented as a existential quantifier at the widest scope, which can be ignored in all cases except scope ambiguity, where we only have indefinitely-quantified NPs. Similarly, Artzi et al. (2015) introduce Skolem terms (Steedman, 2011) for definite NPs, which are also globally scoped. Thus, we do not differentiate between definite and indefinite NPs in AMP. This has no impact on ambiguity.

### A.1.3 EXTENDING TO NEW TEMPLATES

The framework we release allows for the addition of new templates and lexical items. To add a new template, the user specifies a surface-form template and an LF template, and provides the set of lexical items that can be used to fill slots in the templates. The framework enumerates all possible combinations of lexical items which respect the template constraints and produces paired inputs and LFs.

## A.2 ZERO-SHOT PROMPTS

**PP Attachment** For PP attachment, we show an example the main verb being used transitively, the instrumental use of "with", and the accompaniment use of "with".

```
Let's translate what a human user says into what a computer might say.

Human: Galileo saw Mary
Computer: exists a . saw(a) AND agent(a, Galileo) AND patient(a, Mary)

Human: Mary with the camera
Computer: exists x . exists a . camera(x) AND have(a) AND agent(a, Mary) AND patient
    (a, x)

Human: Galileo saw with the camera
Computer: exists x . exists a . camera(x) AND saw(a) AND agent(a, Galileo) AND
    instrument(a, x)

Human: Galileo saw Mary with the camera
Computer:
```

**Conjunctions** For conjunction ambiguities, we include an example of double conjunction (e.g. *and ... and*) and double disjunction (e.g. *or ... or*). The bracketing can vary.

```
Let's translate what a human user says into what a computer might say.

Human: the bird left and walked and ate
Computer: exists x . exists a . exists e . exists i . bird(x) AND ( left(a) AND
    agent(a, x) AND walked(e) AND agent(e, x) ) AND ate(i) AND agent(i, x)

Human: the bird left or walked or ate
Computer: exists x . exists a . exists e . exists i . bird(x) AND ( left(a) AND
    agent(a, x) ) OR ( ( walked(e) AND agent(e, x) ) OR ( ate(i) AND agent(i, x) ) )

Human: the bird left or walked and ate
Computer:
```

**Bound pronouns** For pronoun coreference, we show the transitive verb and each possible subject with the embedded verb separately. We also include an example of a subject with two verbs so that the model sees how to compose two verbs in the same sentence.

```
Let's translate what a human user says into what a computer might say.

Human: the woman saw Marie
Computer: exists x . exists a . woman(x) AND saw(a) AND agent(a, x) AND patient(a,
    Marie)

Human: Marie smiled
Computer: exists a . smiled(a) AND agent(a, Marie)

Human: the woman frowned and smiled
Computer: exists x . exists a . exists e . woman(x) AND frowned(a) AND agent(a, x)
    AND smiled(e) AND agent(e, x)

Human: the woman smiled
Computer: exists x . exists a . woman(x) AND smiled(a) AND agent(a, x)

Human: the woman saw Marie and she smiled
Computer:
```

**Scope** Scope ambiguity prompts include an example of the verb being used transitively, as well an example of universal quantification.

```
Let's translate what a human user says into what a computer might say.

Human: a bird held a sweater
Computer: exists x . exists y . exists a . bird(x) AND sweater(y) AND held(a) AND
    agent(a, x) AND patient(a, y)

Human: each bird
Computer: forall x . bird(x)

Human: each bird held a sweater
Computer:
```

**Inverse scope** Inverse scope prompts include the same information as scope ambiguities but with reversed arguments.

```
Let's translate what a human user says into what a computer might say.

Human: a dog spotted a hat
Computer: exists x . exists y . exists a . dog(x) AND hat(y) AND spotted(a) AND
    agent(a, x) AND patient(a, y)

Human: each hat
Computer: forall x . hat(x)

Human: a dog spotted each hat
Computer:
```

### A.3 CONSTRAINED DECODING

For locally-run models, we use grammar-constrained decoding (Shin et al., 2021; Shin & Van Durme, 2022; Roy et al., 2022) to ensure that the model produces syntactically-correct formulae. During decoding, we use the BenchCLAMP framework (Roy et al., 2022) to restrict the model's output vocabulary according to a context-free grammar, such that the model can only produce strings accepted by the grammar.[9] This allows us to separate the model's semantic performance from its syntactic abilities. We decode with beam search, using a beam of 5.

### A.4 ZERO-SHOT PARSING: QUALITATIVE ANALYSIS

Section 3.1 shows that models typically perform poorly on zero-shot parsing. Given that we use constrained decoding on all open-source models, the errors they make cannot be syntactic in nature, i.e. their outputs are guaranteed to be well-formed FOL expressions. This raises the question of what kinds of errors models are making. Here, we qualitatively analyze model errors. For each ambiguity type, we sample 10 incorrect examples from the Codegen-16B model and classify the errors the model makes.

- **PP attachment:** There are two classes of errors. 9/10 examples have a missing predicate, e.g. `exists v0 . exists v1 . agent(v1, Adele) AND have(v1) AND instrument(v1, v0) AND spied(v1) AND telescope(v0)` is missing a `patient(v1, Sherlock)` predicate for the sentence "Adele spied Sherlock with a telescope". The remaining example had incorrect variable usage.
- **Scope:** 9/10 incorrect examples had a $\exists$ in place of the $\forall$ quantifier (i.e. the right number of quantifiers but no $\forall$ quantifier). 1/10 was missing a $\exists$ quantifier.
- **Inverse Scope:** 10/10 examples had a $\exists$ in place of the $\forall$ quantifier.

---

[9]We release our FOL and Lisp grammars.

- **Bound:** 8/10 examples used the same variable for 2 verbs. For example, in `exists v0 . agent(v0, Katherine) AND frowned(v0) AND observed(v0) AND patient(v0, Mary)`, the event variable `v0` is used for `frowned` and `observed` when there should be an additional event variable `v1`. 1/10 examples had other incorrect variable use, and another had a different missing predicate.

- **Conjunctions:** 8/10 examples had the wrong connectives, i.e. `OR, OR, AND` instead of `OR, AND, AND`. 2/10 had bad scoping, where the correct predicates and connectives were produced but were grouped incorrectly.

## A.5 ANNOTATION TASK

To ensure the validity of our results, we conducted a pilot paraphrasing task, where we asked Mechanical Turk annotators to verify that they were native English speakers and paraphrase a short passage. In the task, annotators from a trusted list were first asked if they were native English speakers. To verify the results, annotators were additionally asked to paraphrase a short fable (The North Wind and the Sun) in 3 sentences. The annotation interface precluded copying, preventing annotators from using external resources. Annotators were paid $0.50 for the pilot task, corresponding to an hourly payment of $\sim \$14.00$ Each annotator's summary was then manually checked to verify fluency and adequacy; all annotators passed the quality check.

In the main HIT, annotators were again paid $\sim \$14.00$ per hour, and each annotator performed exactly 20 annotations in a sequence. Each sequence has 5 tuples of the 4 ambiguity types. Annotators were shown a sliding scale with 3 ticks: *not confident*, *somewhat confident*, *very confident*, and annotators can slide the indicator anywhere along the scale. The interface can be seen in Fig. 7.

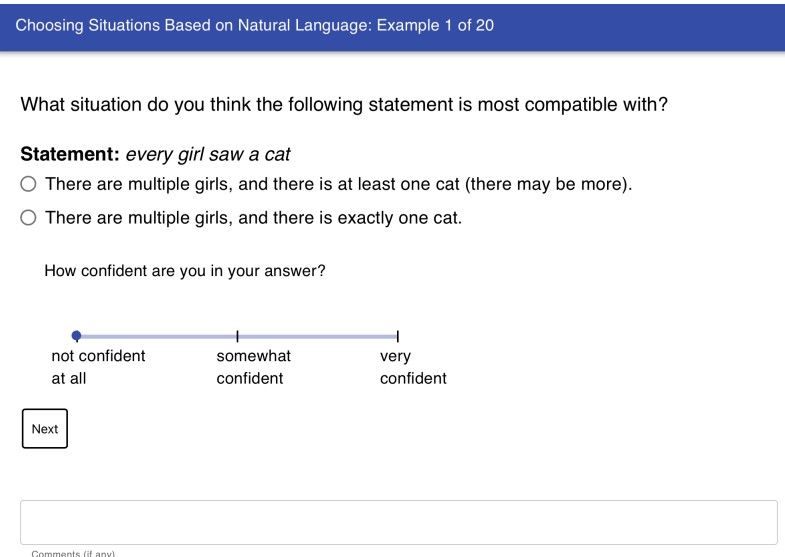

Figure 7: Annotation interface for human evaluation.

To transform the annotators' confidence scores into probabilities, we first min-max normalize raw confidence scores, following past work using sliding bars (Vashishtha et al., 2019). This accounts for the fact that different annotators may use the slider differently. We then take the lowest confidence value to correspond to $p(LF_c) = 0.5$, where $LF_c$ is the LF corresponding to the chosen interpretation. Intuitively, if $p(LF_c)$ were less than 0.5, the annotator would have chosen the other LF. The highest confidence value corresponds to $p(LF_c) = 1.0$.

Qualitatively, we find that visual verbs and nouns (e.g. *saw-telescope*, *observed-glasses*) are matched more to $LF_1$, where the PP is an instrument, while tactile verbs and nouns (e.g. *held-gloves*, *picked up-mittens*) yield a possessive interpretation.

**Limitations of the Annotation Task**   To gather human preferences, we elicit choices between verbalized interpretations of each logical form. This is a different task from what the models are being tasked with, and is motivated by the fact that annotators are unlikely to know first-order logic. Even if they did, it is difficult to constrain annotators to produce exactly the kind of first-order logic statements that would match the reference. In this sense, most of the models we test have an advantage, as we use constrained decoding according to a grammar. Thus, the model cannot produce LFs that deviate from the syntax expected by AMP.

Our method for eliciting judgements differs from standard methods in psycholinguistics, which are typically based on reading times, eye tracking, or other more elaborate experimental methods (Lackner & Garrett, 1972; Rayner et al., 1983; Filik et al., 2004; Dwivedi, 2013). It is more akin to the paraphrase verification method used by Rayner et al. (1983) to elicit interpretations. Note that these methods test for different things. While the former set of methods typically test for incremental and subconscious processes, our method and paraphrase verification test for conscious, non-incremental judgements. In the context of a comparison to transformer-based models which do not receive incremental input, the second paradigm is a more accurate fit. Nevertheless, the uncertainty consciously expressed by our human annotators may differ, for example, from the uncertainty we would obtain via more direct measurements like reading times.

