# OpenReview forum: "Zero and Few-shot Semantic Parsing with Ambiguous Inputs"
_ICLR.cc/2024/Conference — ICLR 2024 poster_

### Official Review · Reviewer_dZKu · 2023-11-01

**Soundness:** 4 excellent
**Presentation:** 4 excellent
**Contribution:** 3 good
**Rating:** 8
**Confidence:** 3

**Summary:**

This paper addresses an important class of tasks: evaluating semantic parsing outside of the realm of 1:1 mappings from form to meaning, potentially including an actual distribution over meanings.   They develop a dataset of templatically generated text-LF pairs where each sentence maps to two possible interpretations, use smart annotation of scalar rankings to get distributions over those ambiguities, and evaluate models on their ability to generate both possible meanings given some prior in-context-learning prompts, and ability to match the distribution of human judgements and to match inbalances of interpretion in the ICL inputs.  They find that models (most of the codegen variety) do quite bad as "zero shot" prediction of these LFs, somewhat well at producing a best LF in certain conditions but also quite poorly at generating well-calibrated distribution over the two choices.

**Strengths:**

- It is a well-written paper with both clear reasoning and clear explanation, enough detail to be replicable and with the promised of publication of data and code.
- It served to target specific questions about how models handle these ambiguities (e.g. the mapping token-level confidence to LF outputs), and is properly planned out to answer those questions.
- Both the human judgments annotation and the constrained decoding seem very rigorously done.

**Weaknesses:**

- While just a quibble, I feel like the combination of ICL and very constrained templates makes their zero-shot setting very hard, and it's hard to draw conclusions from their findings there.

**Questions:**

- In more real-world semantic parsing tasks there can be dramatically more than just two possible semantic parses. Would this approach (particularly use of EASL in collecting human judgements) still work in contexts with far more acceptable parses per sentence?
-  Insofar as all models were equally poor at a number of the zero-shot tasks at ZM_5, did the authors check the effect of raising that k parameter?
- I was curious about the focus on capturing the training data biases rather than the human annotation judgements, for the few-shot setting.

---

> ### Author Response · Authors · 2023-11-17
> **Response to reviewer dZKu**
>
> Thank you for your comments and questions about our work, and for your attention in reviewing it. We've attempted to address the comments and questions from your review below and in the updated paper.
>
> **Weaknesses:**
> - *“I feel like the combination of ICL and very constrained templates makes their zero-shot setting very hard…”*
> Thanks for making this point. The constrained nature of the templates is indeed a limitation of our approach – however, we would like to note that the use of constrained decoding does alleviate this concern somewhat. Because the model is constrained to use the grammar of our templates, it can only generate syntatically-correct outputs. Thus, the model has help in generating valid templates, even if the template format is not what the model would naturally generate during open generation. Nevertheless, the point that open-source models struggle with generating correct LFs of either kind in the 0-shot setting is well-taken, and is an area for further research.
>
> **Questions:**
> - *“In more real-world semantic parsing tasks there can be dramatically more than just two possible semantic parses. Would this approach (particularly use of EASL in collecting human judgements) still work in contexts with far more acceptable parses per sentence?”*
> The human judgment collection would be substantially more difficult given >2 parses, but still possible. We could in principle use the same interface, and change the confidence scale. Right now, 0 confidence is translated to p=0.5 and 1 is mapped to p = 1. The logic behind this is that if the annotator were below p=0.5, they would have chosen the other option with some confidence > 0. This could be extended to multiple options, i.e. given 3 options, 0 confidence would mean p = 0.33. Alternatively, given multiple options, we could apply ranking-based approaches for data collection instead.
> - *“Insofar as all models were equally poor at a number of the zero-shot tasks at ZM_5, did the authors check the effect of raising that k parameter?”*
> Due to computational constraints, we were unable to run higher batch sizes at scale. In our preliminary investigations, for a small subset of examples we found that the models we could run generally have poor beam diversity even with higher beam sizes, generally generating versions of the same output. After a certain point, the models tend to generate outputs that are structurally similar but have incorrect predicates in them, e.g. the predicate boy(x) might be used even though “boy” doesn’t appear in the input. This suggests that increasing the beam size is unlikely to change the ZM scores by very much in practice.
> - *“I was curious about the focus on capturing the training data biases rather than the human annotation judgements, for the few-shot setting.”*
> Thanks for this point, it suggests an interesting direction for future work. In developing the FIM/FDM scores, we did consider whether to compare against human judgements. We opted to instead compare the range of possible ratios for two reasons. Firstly, we found in our human judgements that ambiguity is highly variable between examples, meaning we would need large amounts of data to make an accuracy comparison. For example, in Fig. 6 (now Fig 5.) PP attachment ambiguity examples range from 0 to 1. Averaging this to say that annotators generally think they have P(LF0) = 0.5 would obscure this variation, so we’d need instance-level annotations for all our evaluation examples. Secondly, ambiguity may not be a fixed quantity: we found that annotators differ confidently in their interpretations. Additionally, as pointed out by reviewer PRrS, there might be cultural differences in interpretation. Thus a fixed sample of interpretations might not be valid across different groups of people. We opted for metrics that subsume a metric based on estimates of judgments. Because we range the percentage of examples in the prompt from 0-100% in FDM and FIM, the true interpretation ratio should be found somewhere on this range. For example, if 30% of people interpret an example as LF0 and 70% interpret it as LF1, then this should be captured by FDM/FIM, since they test for this ratio. Marginalizing across ratios allows us to be agnostic about what the “true” ratio might be. Future work could expand this to a finer granularity – we considered increments of 10%, but given more examples we could consider finer increments.

---

> > ### Comment · Reviewer_dZKu · 2023-11-22
> >
> > I thank the authors for their answers to my questions! I'm satisfied with those answers and appreciate the additional detail.  I do  appreciate some of the concerns voiced by other reviewers regarding impact, but it still seems to me to make worthwhile progress on a difficult domain, and so have decided to keep my current score.

---

### Official Review · Reviewer_49nJ · 2023-11-02

**Soundness:** 2 fair
**Presentation:** 3 good
**Contribution:** 2 fair
**Rating:** 6
**Confidence:** 2

**Summary:**

The paper proposed a task to convert an ambiguous sentence to its logical form to evaluate LLMs with zero-shot and few-shot evaluation. The paper studies five types of ambiguity and evaluates LLMs with a decent number of examples. They also proposed multiple metrics to interpret the model results. The dataset and experiment all together provide insights into how LLMs understand ambiguity and its difference from human ambiguity understanding.

**Strengths:**

1. The motivation and writing are very clear. The paper is generally easy to follow.
2. I like the human probability vs. model probability experiments personally, and seeing that humans have certain preferences on one interpretation than the other is interesting, and model prediction somehow matches it as well is very interesting too.

**Weaknesses:**

1. The generation task is hard, especially generating logical forms. Why not formulate this as a multi-choice problem?  Letting the model choose two from 10 possible combinations?
2. Is there any quantitative analysis? What kind of errors does the model usually make?
3. The evaluation metric can be improved. I have several questions about this. Why not use the same zero-shot and few-shot metric since the output format is the same? Why not use language interpretation instead of LF generations? Language interpretation is much more intuitive for the model than LF generations.

**Questions:**

1. Why not report ZM 100 in Figure 4?
2. In section 3.1, the zero-shot experiment, how do you get multiple predictions? Do you sample k times? Your prompt doesn’t encourage the model to predict two LFs, so the model only predicts one by demonstration. The zero-shot results could get better if you encourage two predictions in your prompt.
3. in figure 2, any intuition of why cg-2b is outperforming cg-6b on LF1?
4. It's more like a suggestion: Do you plan to extend the ambiguity to more than two interpretations?

Notes:

- Section 2.1 Metrics. The last sentence is confusing: “The higher, the better. “ If you are talking about k values, shouldn't it be the lower, the better? If it refers to the metric, then it should be explicit.
- Equation 1 is very confusing. Do i  and k mean the same thing here?
- Section 3, paragraph 2, “Here”⇒ “In this setting”
- Figure 4 is a Table, not a Figure.
- Figure 5 is a bit hard to read when it comes to different types of semantic ambiguity. You can choose to put a different text color for important tokens inside the blue circle to increase the readability.
- In Section 4, paragraph 2, what does “1 sentence at a time” mean when the percentage of the LF0 sentences is from 0 to 100 in increments of 10?

---

> ### Author Response · Authors · 2023-11-17
> **Response to reviewer 49nJ (weaknesses)**
>
> Thank you for your attention to our work and your thorough questions and feedback. We have made several changes to the paper, which we hope address your concerns and which we highlight below.
>
> **Weaknesses**
> 1. “...Why not formulate this as a multi-choice problem?...”
> This is a good question, which we have sought to address further based on your feedback. The reason to explore this problem semantic parsing (as opposed to multiple choice or open-ended language generation) is because of semantic parsing’s real-world applications. Parsing is used directly in real-world systems that have real consequences (e.g. robots, digital assistants, etc.). Ideally, these systems would be able to operate in open-ended environments, i.e. environments where we cannot frame the problem as multiple choice QA (since we don’t always know the choices, or there are too many to enumerate). Thus, while we’re interested in model’s performance on ambiguous language generally, we focus specifically on semantic parsing because of its practical applications. We agree that the results in a multiple choice setting may be very different from what we have presented, but these results would have little bearing on semantic parsing. We have added further motivation for exploring semantic parsing specifically in the introduction and in the discussion of implications of our work in the discussion (highlighted in blue).
> 2. *“Is there any quantitative analysis? What kind of errors does the model usually make?”*
> Based on this helpful suggestion, we’ve added a qualitative analysis to the appendix describing typical errors the model makes for each ambiguity type. We find that the best open-source model makes different errors depending on the ambiguity class, but many errors involve omitting key predicates or quantifiers.
> 3.
>     1. *“ Why not use the same zero-shot and few-shot metric since the output format is the same?”*
> The expected/desired behavior in the zero and few-shot settings is different. In the zero-shot setting, we don't know what the "true" output distribution is, so the metric just looks at whether the model puts enough mass on both interpretations to generate them. In the few-shot setting, we're giving the model evidence of the expected distribution, so we have a reference point, and can quantify whether the model captures that distribution. Thus, we need different metrics for the two settings.
>     2. *"Why not use language interpretation instead of LF generations? Language interpretation is much more intuitive for the model than LF generations."*
> This is certainly true. However, in addition to the reason mentioned above for multiple choice (we care about semantic parsing, as opposed to general language generation) evaluation is very hard for language interpretation, and generally requires human evaluation, while automated evaluation for semantic parsing is easier since the form is constrained to a specific language.

---

> > ### Author Response · Authors · 2023-11-17
> > **Response to reviewer 49nJ (questions/typos)**
> >
> > **Questions/Typos/Notes**
> > - *“Why not report ZM 100 in Figure 4?”*
> > Decoding is computationally expensive to run. Typical semantic parsing systems use a beam-size of 5-10 for efficiency, so we follow past work here.
> > - *“…The zero-shot results could get better if you encourage two predictions in your prompt.”*
> > Thanks for raising this point. While it’s true that we could likely perform better on ambiguous examples given a prompt for ambiguity, this prompt might be different from the overall best prompt for unambiguous examples. In that case, we would need some way of choosing the prompt based on the ambiguity in the example. In practice, semantic parsing systems typically use a single prompt for all examples. So here we use the same prompt as other past work ([Shin et al. 2021](https://arxiv.org/abs/2104.08768), [Shin and Van Durme 2022](https://arxiv.org/abs/2112.08696), [Roy et al. 2023](https://openreview.net/forum?id=k4juAEW1tG) )
> > - *“in figure 2, any intuition of why cg-2b is outperforming cg-6b on LF1?”*
> > It’s not entirely clear – qualitatively, we examined the outputs of cg-6B and found that it largely omits the universal quantifier in its output for scope.
> > - *“It's more like a suggestion: Do you plan to extend the ambiguity to more than two interpretations?”*
> > This is a great suggestion. We are in fact working on this for an ongoing project – the AMP code does easily allow for >2 interpretations, though we choose to focus on 2 for the sake of clarity in this work.
> > - *“Section 2.1 Metrics. The last sentence is confusing: “The higher, the better. “ If you are talking about k values, shouldn't it be the lower, the better? If it refers to the metric, then it should be explicit.” and “Equation 1 is very confusing. Do i and k mean the same thing here?”*
> > We have updated the description to clarify these points.
> > - *Section 3, paragraph 2, “Here”⇒ “In this setting” and “Figure 4 is a Table, not a Figure.”*
> > Thanks – we have fixed these issues.
> > - *“Figure 5 is a bit hard to read when it comes to different types of semantic ambiguity. You can choose to put a different text color for important tokens inside the blue circle to increase the readability.”*
> > Thanks for this suggestion. We have updated the text color based on the background color in this figure.
> > - *“In Section 4, paragraph 2, what does “1 sentence at a time” mean when the percentage of the LF0 sentences is from 0 to 100 in increments of 10?”*
> > Thanks for pointing this out – we meant that we vary the number of LF0 sentences in the prompt from 0 to 10, in increments of 1. Since there are 10 sentences total, this is the same as varying the percentage from 0 to 100 in increments of 10. We have updated the section to make this clearer.

---

> > > ### Author Response · Authors · 2023-11-21
> > > **Follow-up on response**
> > >
> > > We would like to follow up to see if our response addresses your concerns (especially those about the framing of the task as semantic parsing rather than multiple choice or open-ended generation) or if you have any further questions. We would greatly appreciate the opportunity to discuss this further if our response has not already addressed your concerns. Thank you again!

---

> > > > ### Comment · Reviewer_49nJ · 2023-11-21
> > > > **Thanks for the answers**
> > > >
> > > > Thank you for the answers, I updated my score.

---

### Official Review · Reviewer_Qhxa · 2023-11-02

**Soundness:** 3 good
**Presentation:** 4 excellent
**Contribution:** 3 good
**Rating:** 6
**Confidence:** 4

**Summary:**

This paper presents a benchmark suite and set of corresponding metrics for measuring models' behavior in the face of ambiguity in a semantic parsing task. The task is set in an in-context learning setting where models are provided with examples of (natural language, logical form) pairs and asked to translate a new natural language sentence into a logical form (LF).

There are two test settings:
* A "zero-shot" setting where the model is expected to infer the existence of an ambiguity in the input example that did not show up in its few-shot examples. For example, by Appendix A.2, the model might see the LF for `Mary with the camera` (which, side note, seems ungrammatical to me at least when interpreted in the relevant sense) and "Galileo saw with the camera" (which...also seems ungrammatical absent context), it's tested on whether it recovers that "Galileo saw Mary with the camera" is ambiguous, by checking if both possible interpretations get high enough probability (or ranking) in its output distribution.
* A "mixed prompting" setting where the model is given ambiguous examples in the prompt which resolve one way or the other a given proportion of the time (e.g., attaching high or low) and it is tested on the degree to which it approximates this ambiguity resolution behavior in its outputs (either on a dataset level in what it ends up decoding on lots of examples, or on an instance level in how it allocates relative probability).

**Strengths:**

* The paper aims at an important problem (handling ambiguity in semantic parsing).
* The setup is clever and allows for some interesting analyses. I think the looking at the token-level confidences to see how model uncertainty is reflected in ambiguity-resolution–dependent choice points, as done in Figure 5, is a useful idea.
* The comparison to human behavior (Section 3.2) is interesting and I imagine could seed future experiments.

**Weaknesses:**

I'm worried about how we assign meaning to the various results, and I'm not sure how this result would feed into future work that helps parsers handle ambiguity better.

1. Human experiments: humans were given both interpretations and asked to assign confidences to them. This seems a bit different from what the models were asked to do in the zero-shot experiments, which is implicitly pick out the ambiguity on their own. I understand it'd be hard to elicit this kind of behavior from humans — ideally you would ask them a question whose answer hinges on the ambiguity and get some measure of the uncertainty in their answer, but it's unclear what that measure would be. The measured degree of uncertainty from humans in this setting may quite exceed the actual uncertainty when reading and interpreting language in practice, as humans may simply snap to one reading or the other. So it may not be fair to compare this result to model results — though of course, there is no clear analogy between what humans and models are doing anyway, so I'm not sure what the exact criteria for a fair comparison should be. It might seem like a safer bet to compare humans to models on a similar task, where the model is given both interpretations and is asked to give a confidence score, or repeat the one that it prefers — for example, I think it'd be a strict improvement on the current paper to do the human comparison to the FIM relative confidence results instead. But of course, the confidences reported through that process may not be ecologically valid with respect to the paper's broader goal of characterizing LM behavior on ambiguous semantic parsing examples.
2. More broadly, the paper relies on a mostly implicit assumption that the _appropriate_ behavior for an LM is to represent multiple possible readings in its top-k outputs, or for the probabilities of each possible reading to reflect the proportions of related readings in their prompts / ICL data. It isn't obvious to me that this is what we want or that it is the right way of approaching the question of whether LMs can handle ambiguity in semantic parsing appropriately. When prompted for a single output, why _should_ the model distribute its probabilities in any particular way? There's nothing that says that we have to use the top-k candidates in order to represent a set of semantically distinct alternatives, and indeed it is suboptimal for this purpose, as the paper notes that the top-k results are usually variations of the same interpretation. I can see how this may make sense if we were to, for example, use the LM probabilities as a prior to guide some kind of search through the space of parses; we then want all plausible interpretations to be discoverable. But it's not clear that's the best strategy in practice if we're going to be using LMs anyway: why not just ask it, for example, to decode all possible interpretations in sequence? There's a big space of possible things to do here if we want models to handle ambiguity, and unfortunately the results of these experiments only bear on a small subset whose promise is unclear. Not that the experiments done are bad or invalid, but I think the paper needs to make a clear argument about the nature of the construct being tested and what these results are supposed to inform.
3. On that note, it's unclear to me from the arguments in the paper how these results can actively guide future efforts to build semantic parsing systems that handle ambiguity better. I think the burden does lie on this paper to make that case, at least in principle (I'm not saying it needs more ML experiments).

**Questions:**

Section 3.1:
* I think Figure 2 would be better as a grouped bar chart? It doesn't really make sense to me to have multiple models along a line here.
* "we rarely observe any models predicting both LFs correctly; one exception to this is conjunction ambiguity" — seems also true for `bound`?

Section 6:
* "statements be ambiguous" (typo)
* "Making this assumption can reduce semantic parsing to syntactic parsing" — it's very unclear to me what this means
* The argument in the second to last paragraph that datasets need to have multiple judgments seems wrong to me. If there is indeed annotator disagreement, this disagreement will show up implicitly spread across multiple examples. Learning to maximize the likelihood of the data should teach the model the right kind of uncertainty. Multiple annotations aren't actually necessary for learning this — though they might be necessary for _evaluating_ it.

---

**EDIT:** Thanks to the authors for their response. I think all of my concerns were basically addressed except for the point about the choice to ask that the model represent ambiguity in its top-k probabilities, which I think is a problematic assumption (outlined why in my comment). I think this undermines the significance of the paper as it takes an outdated and limiting perspective on how to represent ambiguity, but I still think it constitutes a useful contribution to the literature, hence I'm sticking with my recommendation of a weak accept.

---

> ### Author Response · Authors · 2023-11-17
> **Response to Qhxa**
>
> Thank you for your thorough and insightful comments and questions. We have attempted to address most of the points made in our uploaded draft, where edits can be seen in blue. Below, we address each point in more detail.
>
> **Weaknesses**
> - *“Human experiments…”*
>
> We are in agreement that the human results are imperfect, and have added a section on limitations of our human judgements to the appendix contrasting our study with standard methods in psycholinguistics and providing further justification for our choices. Regarding a comparison between human judgments to FIM: as-is, this judgment would not be a completely fair comparison, since human annotators were not shown any examples, while the model is shown 10 ICL examples with a particular ratio of LF1 to LF2.
> - *Assumptions about desired behavior*
>
> Thank you for pointing out this assumption. We do state the desired behavior of the model at the beginning of paragraph 3, but have added clarification about what the actual “model” is. As you astutely put it, having a distribution over interpretations makes sense if we are to “use the LM probabilities as a prior to guide some kind of search through the space of parses”. Since actual generations are being produced by beam search – which exactly is a search through the space of parses, guided by the LM’s probabilities – I think that we are in fact in agreement on the importance of the probabilities. We have added a point to introduction, clarifying that what is considered the “parser” is not only the LLM but also the search procedure used to actually decode parses (in our case, beam search). In terms of why we use beam search rather than sequentially decoding parses: we are aiming here to remain fairly close to how semantic parsing models are used in practice, rather than optimizing for performance on ambiguous examples specifically. We have clarified this in our limitations section.
> - *Guiding future research*
>
> Thanks for this feedback, which is shared with Q1 from reviewer PRrS. In light of this, we have updated the discussion section to include more direct recommendations for how we can improve semantic parsing under ambiguity (and why we might want to).
>
> **Questions/Typos**
> - *“I think Figure 2 would be better as a grouped bar chart? It doesn't really make sense to me to have multiple models along a line here.”*
> Based on your recommendation, we tried this plot style. We found that it was hard to parse which ambiguity type belonged to which model, since many of them (in fig. 2) are 0, so there are large gaps in the bar plot. These make it hard to determine which bars correspond to which model, which is clearer from the line plot.
> - *“"we rarely observe any models predicting both LFs correctly; one exception to this is conjunction ambiguity"— seems also true for bound?”*
> Thanks for pointing this out – we have amended the text accordingly.
> - We have clarified the point about semantic parsing being reduced to syntactic parsing. If there is a 1-to-1 mapping between syntax and meaning, then semantic parsing can be handled by a syntactic parser (this is of course not the case for real data).
> - Re. the point on “multiple judgments”: Thank you for this observation, it is well-taken. We have amended the section to clarify that redundant annotations would only be required for the evaluation data, but that the train data requires diverse annotations. (see the added footnote 8)

---

> > ### Author Response · Authors · 2023-11-21
> > **Follow-up on response**
> >
> > We would like to follow up to see if our response addresses your concerns about the human experiments and the clarity of our assumptions, or if you have any further questions. We would really appreciate the opportunity to discuss this further if our response has not already addressed your concerns. Thank you again!

---

### Official Review · Reviewer_PRrS · 2023-11-03

**Soundness:** 4 excellent
**Presentation:** 4 excellent
**Contribution:** 4 excellent
**Rating:** 8
**Confidence:** 4

**Summary:**

The paper explores the longstanding issue of ambiguity in natural language and its implications for semantic parsing in AI. It delves into the nature of language ambiguity, highlighting how it stems from the balance of communication efficiency and interpretive flexibility, and poses challenges for AI systems that lack human-like commonsense knowledge and context. To address these challenges, the authors propose a novel framework and the Ambiguous Parsing (AMP) dataset, which includes various types of ambiguities paired with dual logical forms (LFs). This resource is aimed at enhancing the performance of large language models (LLMs) in semantic parsing tasks, especially in handling ambiguity.

The study introduces two tasks to assess how well LLMs, utilizing in-context learning (ICL), can capture multiple interpretations of an ambiguous input. These tasks are designed to evaluate model performance in both zero-shot and few-shot settings, with a series of metrics developed to quantify their ability to predict and represent ambiguity. The paper also reports on models' performance, noting that while models can sometimes mirror human preference for certain interpretations, they generally fall short in predicting all possible parses. Additionally, it is observed that some models are quite adept at reflecting the distribution of interpretations in mixed-prompt scenarios, offering insight into in-context learning amidst conflicting evidence.

**Strengths:**

1. The AMP dataset is a significant contribution, providing a resource specifically designed for investigating ambiguity in semantic parsing, which is a relatively unexplored area.

2. The paper takes a comprehensive approach by addressing the challenge from the perspective of both dataset creation and model evaluation.

3. The introduction of zero-shot and few-shot tasks offers a rigorous evaluation framework for future research on ambiguity in semantic parsing.

4. The development of new metrics to assess the models’ ability to handle ambiguity is a noteworthy contribution that can guide subsequent model development.

5. The results contribute interesting insights into the capabilities and limitations of current LLMs in capturing ambiguity through zero-shot and in-context learning.

**Weaknesses:**

1. While the paper provides a strong foundation, it could benefit from a more detailed exploration of how ambiguity affects real-world applications of semantic parsing.

2. The AMP dataset, while novel, might still be limited in scope and diversity, potentially affecting the robustness of the study’s conclusions.

3. It is unclear how the proposed methods deal with the dynamic nature of conversational context, which can significantly affect ambiguity resolution.

**Questions:**

1. How do you foresee the findings of this research being applied in practical AI systems, particularly in areas where ambiguity can have significant consequences, like in legal or healthcare settings?

2. Is the AMP dataset extensible, and are there plans to include more complex or nuanced forms of ambiguities, such as cultural or idiomatic ones?

3/ Could you elaborate on the selection process for the five types of natural language ambiguities included in your study? Were there other types of ambiguities considered but excluded?

4. The use of synthetic data might not fully capture the complexity of natural language ambiguities encountered in real-world scenarios. How well do the findings translate to naturally occurring datasets?

---

> ### Author Response · Authors · 2023-11-17
> **Response to reviewer PRrS (weaknesses)**
>
> We would like to thank the reviewer for their thorough review and attention to our work. We have uploaded a new draft of the paper (revisions in blue) where we have sought to address the weaknesses pointed out. We describe our changes/reasoning in more detail below:
>
> **Weaknesses**
> 1. *“[the paper] could benefit from more detailed explanation of how ambiguity affects real-world applications of semantic parsing.”*
> We have added more details to the introduction, motivating why we examine semantic parsing and how misunderstandings could arise from ambiguity.
> 2. *“The AMP dataset, while novel, might still be limited in scope and diversity, potentially affecting the robustness of the study’s conclusions.”*
> This point is well-taken – while our study is an initial exploration into the underexplored area of ambiguity in semantic parsing, it is far from complete in terms of the range of possible ambiguities (and languages, settings, cultures, etc.) that could be explored. We have clarified our limitations subsection in Section 6 to emphasize that AMP can be extended both structurally (covering scope) and lexically (covering diversity).
> 3. *“It is unclear how the proposed methods deal with the dynamic nature of conversational context, which can significantly affect ambiguity resolution.”*
> Thank you for pointing this out – this is definitely a limitation of our approach, since conversational context often reduces ambiguity. We have highlighted where we state this assumption in the intro and have added it to the limitations. Removing context is realistic in the “digital assistant” domain, since most people would not accept a digital assistant that is constantly listening, i.e. by design, assistants have access to only parts of conversations. Note that some ambiguity is likely to persist even with context, as ambiguity commonly leads to breakdown in human conversation.

---

> > ### Author Response · Authors · 2023-11-17
> > **Response to reviewer PRrS (questions)**
> >
> > **Questions:**
> >
> > Thanks for your questions -- we have one follow-up question under Q2.
> >
> > 1. *“How do you foresee the findings of this research being applied in practical AI systems, particularly in areas where ambiguity can have significant consequences, like in legal or healthcare settings?”*
> > Our work has several practical implications. First, the finding that models seem to capture ambiguity poorly in zero-shot settings has implications for safety. Given that people tend to disagree confidently about ambiguous utterances, we believe that people might give ambiguous instructions without recognizing their ambiguity. If the model parses these differently from how they were intended, it could lead to unintended consequences across many domains (healthcare, legal applications, digital assistants, physical robots). For example, in the healthcare setting, an ambiguous sentence like "the patient presents with a cough and fever or a runny nose" might lead to an incorrect diagnosis (i.e. if the patient has just a cough). Secondly, the finding that models capture the distribution in the prompt is more promising. It suggests that, given evidence of ambiguity in the training data, we can adequately capture the distribution of human judgments. This in turn suggests that we need to collect data on ambiguity and obtain judgements from multiple people. We have added details to the intro and discussion to make these points clearer.
> >
> > 2. *“Is the AMP dataset extensible, and are there plans to include more complex or nuanced forms of ambiguities, such as cultural or idiomatic ones?”*
> > AMP is easily extensible – additional templates and vocabulary items can be added, and the code offers instructions on how to do so (see updated footnote 2). The suggestion to add cultural or idiomatic ambiguities sounds extremely interesting – could you give an example of a cultural ambiguity? Would this be an utterance that could be interpreted one way by a listener from one culture, and another way by a listener of another culture?
> >
> > 3. *“Could you elaborate on the selection process for the five types of natural language ambiguities included in your study? Were there other types of ambiguities considered but excluded?”*
> > We selected our ambiguity types by reviewing the relevant literature in linguistics/psycholinguistics and pulling out well-documented and researched ambiguities. We did exclude lexical ambiguities from our study. Our rationale for this was that LLM models are fairly adept at word-sense disambiguation tasks, which have long been studied in NLP. To make examples non-trivial, we would have to come up with very contrived examples. Additionally, we found that lexical ambiguities only occur in very specific contexts, making generating data for them challenging. For example, a standard lexical ambiguity is “bank (monetary institution)” vs “bank (of a river)”. Some sentences allow for both (e.g. “I went to the bank” is ambiguous) but we couldn’t drop in another ambiguous noun, e.g. “rock (geology)” and “rock (music)” wouldn’t work in “I went to the rock”. In other words, each lexical ambiguity would need its own specific example, which is already covered by existing word-sense disambiguation data.
> >
> > 4. *“The use of synthetic data might not fully capture the complexity of natural language ambiguities encountered in real-world scenarios. How well do the findings translate to naturally occurring datasets?”*
> > Because of a lack of real-world data on ambiguity, we don’t have quantitative results here. Speculatively, because we're using LLMs and 0-shot learning, the sim-to-real gap between synthetic and real data might be smaller. Normally, the problem with using synthetic data is that models trained on synthetic data don't generalize to real data. In our case, we don't train any models, and instead test a model trained on real data on synthetic data. While the results might differ, we would expect that (given that the model was trained mostly on real data) if it generalizes to synthetic data, it should also be able to generalize to real data, which is actually closer to the model's data distribution.

---

### Author Response · Authors · 2023-11-17
**Overall response to reviewers**

We would like to thank all of the reviewers for their attentive and thorough reviews. We have made several changes to address the reviewers’ concerns and questions. At a high level, we have further motivated our focus on ambiguity in semantic parsing, as opposed to ambiguity in other settings, and have provided more real-world examples of impacts our work might have on semantic parsing. We have clarified our metrics and the settings we are testing in the few-shot experiments. We have clarified our assumptions and added them to our limitations section, including a new limitations section in the appendix regarding the human experiments. Finally, we have added a qualitative analysis to the appendix. We have highlighted major changes in blue in the text. More detailed responses to each reviewer’s comments can be found below.

---

### Meta-Review · Area_Chair_sJTC · 2023-12-05

**Metareview:**

The explicit study of ambiguity and its representation in the data is interesting, and provides good food for thought.

**Justification For Why Not Higher Score:**

All reviewers identify limitations to how the problem is posed. The AC would like to add two additional issues:

- It would be more interesting of the paper used Python syntax rather than FOL, or in addition. Python is more common in pre-training data, likely making LLMs perform better on it.
- The argument the author provide on synthetic data is not convincing. The dataset can shed some light on the problem of ambiguity, but with no real language (and no account for conversational aspect), the implications of findings is limited.

**Justification For Why Not Lower Score:**

While all reviewers see issues with the paper, there's consensus that the paper makes an interesting attempt at a challenging problem.

---

### Decision · Program_Chairs · 2024-01-16

Accept (poster)